# Taste Beats Reputation in New Food Products Choice: The Case of Ready-to-Eat Pomegranate among Young Consumers in Veneto Region (Italy)

**Alice Stiletto, Erika Rozzanigo, Elisa Giampietri**  **and Samuele Trestini \***

TeSAF Department, University of Padova, 35020 Legnaro, PD, Italy; alice.stiletto@unipd.it (A.S.); erika.rozzanigo@unipd.it (E.R.); elisa.giampietri@unipd.it (E.G.)
**\*** Correspondence: samuele.trestini@unipd.it; Tel.: +39-(0)49-82737

**Abstract:** This study investigates the preferences for ready-to-eat pomegranate arils in Italy through a discrete choice experiment (DCE) on 264 young consumers in Italy. The aim is to estimate consumers' willingness to pay (WTP) for the reputational attributes of the product (e.g., the product origin and sales channel) and to discriminate the elicited preferences between tasting and non-tasting situations. To this purpose, a random parameter logit model was employed to assess the heterogeneity in consumer preferences. The results suggest that non-tasters attach a relevant value to the reputational attributes (e.g., +75% WTP for Italian origin). Moreover, considering the sensory features of the products, we found that consumers in this group discriminate against the proposed samples only through their visual characteristics: they prefer the sample with the largest size and red colored arils. In addition, we found that the tasting experience reduced the value attached to the reputational attributes (e.g., −50% WTP for local origin) for consumers, compared to non-tasting situation, thus shifting their preference to the samples that they appreciated the most (high liking). Specifically, we found that consumers in the tasting group preferred the product sample with the highest level of sweetness and the lowest level of sourness and astringency, showing a higher preference for sweetness. The findings contribute to the literature on consumers' behavior on new food products (NFPs), showing that reputational attributes lose value after the tasting experience. In contrast, the sensory features of the NFPs can help tasters to reduce the information asymmetry, which traditionally represents a hurdle in purchases for new consumers. However, this depends on the individuals' subjective preferences, as demonstrated by the significant effect of liking levels in discriminating consumers' choices. To conclude, although these results cannot be extended to the general population, they may give some interesting insights about future trends of NFP demand.

**Keywords:** discrete choice experiment; random parameter logit; pomegranate arils; NFP; consumer preferences; taste; reputation

## 1. Introduction

Fuller [1] defined new food products (NFPs) as groceries that had not been presented before in any marketplace or as existing products introduced into a geographically new area or with a new package or format. In this context, pomegranate could be considered an NFP in Italy, since its cultivation has a limited (but growing) diffusion [2] and new consumption patterns of this fruit, such as ready-to-eat arils [3], have recently been spreading in the market. In the last two decades, Europe increased the import of pomegranate to about 100,000 tons, for a total value of EUR 109 million/year [4]. This increase is consistent with the growing demand for fresh food with high nutritional value and genuine taste, the so-called superfruit [5–7]. Despite this, consumer preferences for pomegranate fruit remain barely investigated by the literature.

Nowadays, health represents a major driver in the global food market [8], and pomegranate plays a leading role in this context [9–11], due to its huge antioxidant proper-

ties [12,13]. Indeed, health and nutritional benefits are leading factors in purchasing choices especially for "future-oriented" [13] and young [14] consumers, and this is particularly true for pomegranate. One of the latest researches on this fruit [2], conducted in Italy in 2020, found that young consumers are the most prone to buy ready-to-eat arils, since they are the most inclined to try innovative products [15] and to adopt new consumption patterns [16].

However, recent studies have suggested that nowadays consumers are less inclined to renounce an item's taste to have guaranteed positive effects on health [17]. In fact, appreciating the taste of a given product is a key factor in buying choices, as it is unlikely that a consumer will purchase a good for the second time if it did not satisfy his/her hedonic expectations [18]. As pomegranate arils are a novelty in the Italian market, studying the effect of taste on product attributes is of primary importance. Several authors [18–21] stated that, in the hypothetical and non-hypothetical market analysis, taste should be considered when studying consumers' WTP for NFPs. According to Grunert [22], experience attributes as taste can become crucial elements for potential future purchases.

Although numerous studies have considered the role of taste on consumers' propensity to pay a premium price, only few studies combined DCE methodology with sensory and hedonic evaluation. For instance, Baba et al. [23] found that consumers' preferences for beef meat attributes were different before and after tasting the product: reputational attributes as origin were less important after tasting, compared to the meat color. Indeed, due to the imperfect information setting of purchasing decision [24] when deciding what to buy, reputational attributes represent an important factor that need to be considered [25]. In support of this, Barnes et al. [26] found that cheese brands with a high reputation increased consumers' WTP more than taste. However, several studies [27–29] found that the liking for the tasted product could play a decisive role in increasing consumers' WTP and re-purchasing decisions, particularly for new products.

So far, the literature on NFPs has made little effort to investigate the role played by reputational attributes linked to taste experience in purchasing decisions [18,19]. Furthermore, the hypothetical nature of DCE can be overcome when the experiment is performed with real products [30]. In line with this, the present study contributes to the literature on NFP demand by investigating young consumers' preferences and WTP for reputational attributes of pomegranate arils and discriminating the elicited preferences between non-tasting and tasting situation, using real products.

The paper is organized as follows: the next section describes data collection and model specification. Results are reported and discussed in Section 3. Finally, conclusions are drawn in the last section.

## 2. Data and Methods

### 2.1. Experimental Design

This research was carried out in Italy, specifically in Veneto region (north-east of Italy), between November 2019 and January 2020, using commercial pomegranate samples retrieved from the Italian market. The first sample (*HICAZ*) is the most imported variety (*Hicaz*) of Turkish origin [4]. The second sample (*WONS_S*) is represented by a Wonderful variety produced in the south of Italy (Sicily region). The third sample (*WOND_V*) belongs to the same variety of *WONS_S* and it was produced in Veneto region. The chemical and panel characterization of these samples is provided in Rozzanigo et al. [31].

The total sample consisted of 264 consumers, randomly divided into two different groups (Figure 1). The first group (*n* = 132), namely the "no tasting group", followed the standard discrete choice experiment (DCE) approach. Consumers in the first group could evaluate the products considering only the visual and verbal description of their attributes before expressing their choices. The second group (*n* = 132), the so-called "tasting group", tasted and evaluated the overall liking of the three samples on a 9-point hedonic scale before completing the choice experiment. The study was conducted in according to the principles stated in the Declaration of Helsinki (In line with the Declaration of Helsinki, the collected data are completely anonymous, with no personal information being collected.

Data are not sensitive or confidential and the issues being researched do not upset or disturb participants. Moreover, vulnerable or dependent groups are not included and there is no risk of possible disclosures or reporting obligations).

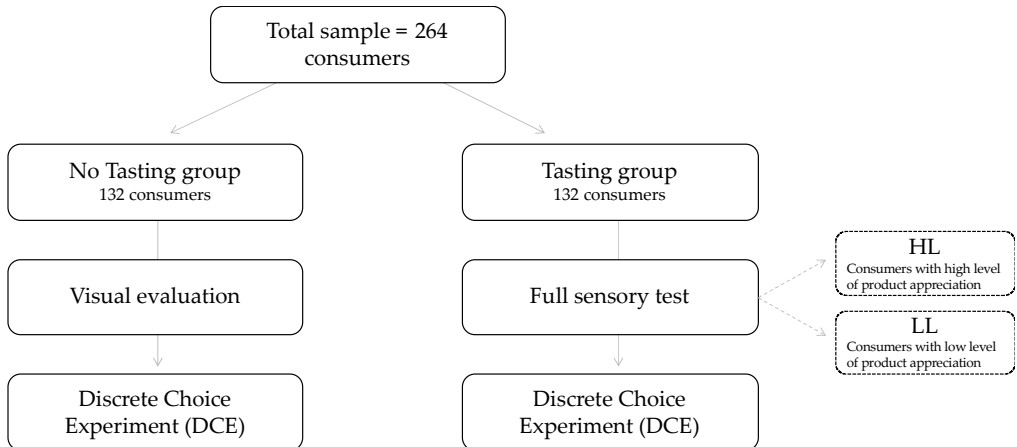

**Figure 1.** Experimental structure. Note: dashed rectangles represent groups defined during the data processing. HL refers to consumers in the tasting group who gave an overall liking score higher than 5 to the product sample tasted. LL refers consumers in the tasting group who gave an overall liking score equal or lower than 5 to the product sample tasted. Source: our elaboration.

The questionnaire was organized in different sections: the first one contained the DCE and the second one was focused on determining consumers' consumption habits related to the pomegranate fruit. Finally, the last section collected information about sociodemographic characteristics of the sample.

Implementing the tasting to the standard DCE design allows to assess the effect of the hedonic score (as obtained by the overall liking of the three varieties during the tasting experience) on product demand to be evaluated [32]. Only during the data processing, within the tasting group we created two dummy variables (HL and LL) to distinguish between consumers who appreciated the product samples that they tasted (HL), and those who did not (LL) (dashed rectangles in Figure 1). It follows that the whole sample (*n* = 264) was divided into the following three groups:

- No tasting group (*n* = 132): consumers that evaluate the product sample only visually;
- Tasting group with high level of product appreciation (HL): consumers who gave an overall liking score higher than 5 (on a 9-point hedonic scale) to the product sample in the choice alternative;
- Tasting group with low level of product appreciation (LL): consumers who assigned an overall liking score equal or lower than 5 (on a 9-pint hedonic scale) to the product sample in the choice alternative.

These dummy variables (HL and LL) interacted with the DCE variables, i.e., the pomegranate's attributes, allowing to estimate the effects of both tasting and liking on the individual preferences to be estimated.

### 2.2. Tasting and Non-Tasting Consumers' Evaluation

Only consumers in the tasting group (132 consumers) performed a sensory evaluation before filling the questionnaire (DCE). Consumers were asked to evaluate the proposed product sample in terms of the overall visual appearance, taste, and texture. More specifically, about 20 arils, manually extracted one hour before the test, were served into odor-free plastic cups with 80 mL capacity, coded with three-digit random numbers. The samples (*WOND_V*, *WOND_S*, and *HICAZ*) were randomly assigned to consumers. The sensory evaluation was assessed using 9-point Just-About-Right (JAR) scales together with 9-point hedonic scales. These scales are often combined to provide directional information, e.g., for

the product reformulation [33]. JAR scale was used for sweet and sour taste, ranging from 1 (e.g., "not sweet enough") to 9 (e.g., "too sweet"), passing through 5, which is the just about right value (i.e., the right intensity of the attribute). The overall visual appearance (color and size) and the texture traits (firmness, juiciness, and seed intrusiveness) were instead evaluated on a classical 9-point hedonic scale, ranging from 1 (totally negative, e.g., "unattractive") to 9 (totally positive, e.g., "attractive"). Finally, the overall liking was evaluated on a 9-point hedonic scale, from 1 ("I dislike extremely") to 9 ("I like extremely").

The respondents tasted and sensory evaluated the three samples before filling in the survey, in line with previous studies [23,26,27,29,32]. (The commercial names of the selected varieties were not indicated either during the tasting or within the DCE, meaning that none of the 264 consumers knew which varieties were objects of study.) Finally, respondents could also re-test the products during the DCE if necessary, but none asked for it.

### 2.3. Choice Experiment

The DCE method is consistent with Lancaster theory [34] of consumer demand and the random utility theory [35]. The first postulates that the utility that a consumer derives from buying a product is not related to the product itself, but to its attributes [33]. According to the second [33], the utility that an individual $n$ derives from choosing the alternative $j$, considering the choice occasion $t$, is given by:

$$U_{njt} = \beta \prime x_{njt} + \varepsilon_{njt} \tag{1}$$

where $x_{njt}$ is a vector of factors observed by the researcher and $\varepsilon_{njt}$ is a vector of unobserved factors.

To assess the consumer preferences expressed through the DCE and considering their heterogeneity, the random parameter logit (RPL) model was implemented [36], for which the utility function is:

$$U_{njt} = \beta'_n x_{njt} + \varepsilon_{njt} \tag{2}$$

where $\beta_n$ is a vector of coefficients specific of the individual $n$ and $x_{njt}$ is a vector of observed attributes that are related both to the individual $n$ and to the alternative $j$ on the choice occasion $t$. Given the $\beta'_n$ and $x_{njt}$ vectors in (2), the probability estimated through RPL model, conditional on knowing $\beta$, can be expressed as:

$$P_{nit}(\beta_n) = \frac{\exp(\beta'_n x_{nit})}{\sum_{j=1}^{J} \exp(\beta'_n x_{njt})} \tag{3}$$

However, the researcher does not know $\beta_n$ and therefore cannot condition on $\beta$ [37]. The density of $\beta$ is denoted as $f(\beta|\theta)$, where $\theta$ are the parameters of the distribution [37]. The unconditional probability of the observed sequence of choices is the conditional probability (3) integrated over the distribution of $\beta$ on $\beta_n$ (i.e., the random parameter logit probability) (4), and it can be expressed as:

$$P_{ni} = \int \frac{exp(\beta' x_{ni})}{\sum_j exp(\beta' x_{nj})} f(\beta|\theta) d\beta \tag{4}$$

### 2.4. Model Specification

DCE was performed both in the tasting and no tasting groups (total number of consumers = 264). The attributes were selected based on the literature on pomegranate and on a preliminary analysis, with open ended questions, which was performed on 32 consumers. The selected attributes of the final choice experiment are the following: sample, sales channel, origin, and price (Table 1). The sample variable refers to the three pomegranate varieties: *WOND_V* (Wonderful variety from Veneto region); *WOND_S* (Wonderful variety from Sicily region) and *HICAZ* (Hicaz variety from Turkey). Consumers in the tasting group were allowed to taste the arils (this simulated a previous consumption experience),

while consumers in the no tasting group could only observe the arils (this simulated a normal purchase of ready-to-eat products on the shelf). Presenting to consumers the arils in odorless and colorless 80 mL cups allows a real ready-to-eat arils purchase to be simulated, considering the similarity with the arils packs currently observable in the market. The sales channel variable refers to the different places of purchase. Results of the preliminary survey suggested that consumers from Veneto region usually buy the fruit not only at the supermarket, but also directly from the farmer, considering that pomegranate is a traditional crop in this area. For this reason, we chose to implement "supermarket", "specialized shop", and "directly from the farmer" levels to the sales channel attribute in DCE. The origin attribute indicates whether the product comes from Italy, or it is imported from abroad, or it is retrieved from local producers (i.e., it derives from Veneto region, considering that the DCE was administered in this region). Finally, as regards the price attribute, three levels were selected based on the current market prices and estimated prices retrieved from both the supermarket and online [38].

**Table 1.** Description of DCE attributes and Levels.

| Attributes | Type | Levels | Codes |
|---|---|---|---|
| *Sample* | Categorical | Wonderful from Veneto<br>Wonderful from Sicily<br>Hicaz from Turkey | *WOND_V*<br>*WOND_S*<br>*HICAZ* |
| *Sales channel* | Categorical | Supermarket<br>Specialized shop<br>Directly from the farmers | *Super*<br>*Spec*<br>*Farmers* |
| *Origin* | Categorical | Local<br>Italy<br>Other than Italy | *Local*<br>*Ita*<br>*Other than Italy* |
| *Price* | Continuous | EUR 1.57/pack of 100 g of arils<br>EUR 2.27/pack of 100 g of arils<br>EUR 2.97/pack of 100 g of arils | |

The full factorial design, namely the set of all possible combinations between attributes and levels, includes 81 alternatives, which is unwieldy. As suggested by Zwerina et al. [39], we reduced the number of possible combinations using a fractional factorial design [40]. This statistical procedure, implemented using IBM SPSS 26.0, allows a fractional factorial to be generated, composed of the main effects without the interactions [41]. The orthogonality of the attributes was maintained during the reduction processes of the combinations [42]. Sixteen choice sets were generated, guarantying the balance level of the attributes. The final number of choice sets were then divided into two blocks (i.e., 8 choice sets per respondent), considering that the number of choice tasks evaluated by the interviewees are generally set between 4 and 10 choice cards [42]. This allows consumers to correctly evaluate the alternatives, avoiding an excessive effort to respondents.

Each respondent had to perform 8 choice sets and was randomly assigned to block 1 or 2. In each choice task, respondents were asked to buy a package of pomegranate arils (100 g) and to indicate their preferences between two multi-attributes alternatives (A and B). These alternatives were different from each other in terms of levels for each attribute. Consumers could also choose not to buy the products (alternative C), if the products described in alternatives A and B did not satisfy them. According to Hensher et al. [43], implementing the "no choice" option allows to replicate a real purchase scenario, in which consumers are not forced to buy a good that does not satisfy them.

Given this framework, the following consumers utility function (5) is used to estimate consumers' choices:

$$U(X) = \sum \beta_i \cdot A_i + \sum \beta_{HLi} \cdot A_i \cdot HighL + \sum \beta_{LLi} \cdot A_i \cdot LowL + \beta_{price} \cdot price \qquad (5)$$

where *A* represents a vector of all attributes apart from price, *HighL* is a dummy variable assuming value 1 if the respondents in the tasting group gave an overall liking score higher than 5 to the product sample tasted, and *LowL* is a dummy variable assuming value 1 if the respondents in the tasting group gave an overall liking score equal or lower than 5 to the product sample tasted. The model was estimated with STATA 16.

## 3. Results and Discussion

### 3.1. Product Characterization

To support the discussion of the results on consumers' preferences (Section 3.2), in Table 2 the physicochemical characterization and panel test on the three samples [31] is concisely reported. Samples are ranked according to the intensity of each attribute for each variety. Specifically, the sample with the highest value for an attribute takes the highest score (i.e., +++), while the sample with the lowest value for that attribute takes the lowest score (i.e., +).

**Table 2.** Summary evaluation of sample characterization through chemical analysis and panel test (ANOVA).

| Attributes | Type of Analysis | WOND_V | HICAZ | WOND_S | F-Test *p*-Value |
|---|---|---|---|---|---|
| *Arils' size* | Physical | ++ | + | +++ | <0.01 (arils length) <0.01 (arils width) |
| | Panel | + | ++ | +++ | <0.01 |
| *Arils' color* | Chemical | ++ | +++ | + | <0.01 |
| | Panel | +++ | + | ++ | <0.01 |
| *Sourness* | Chemical | + | ++ | +++ | - |
| | Panel | +++ | ++ | + | 0.01 |
| *Sweetness* | Chemical | + | ++ | +++ | - |
| | Panel | + | ++ | +++ | 0.41 |
| *Hardness* | Physical | +++ | + | ++ | 0.10 |
| | Panel | +++ | + | ++ | 0.06 |
| *Seed intrusiveness* | Physical | +++ | ++ | + | 0.03 |
| | Panel | +++ | ++ | + | <0.01 |
| *Bitterness* | Chemical | ++ | + | +++ | 0.01 |
| | Panel | +++ | ++ | ++ | 0.01 |
| *Astringency* | Chemical | + | ++ | +++ | <0.01 |
| | Panel | +++ | ++ | + | 0.10 |

Note: +++ indicates the highest score for a given attribute; + indicates the lowest score for it. Sourness and sweetness values derive from only one measure. Source: our elaboration based on Rozzanigo et al. [31].

According to the panel test, *WOND_S* is the sweetest cultivar, and it shows the lowest value of astringency and sourness perceived. Moreover, it has the lowest value in terms of seed intrusiveness.

Focusing on the visual appearance, it emerged that *HICAZ* is the variety that presents the best compromise between color and size, as it was the sample perceived as the most colored and with a medium size of the arils.

### 3.2. Sample Characteristics and Consumption Habits

As shown in Table 3, the sample consists of 132 men and 132 women, with a mean age of 24.08 ± 3.31 years. On average, about 67% of the sample has an upper secondary school education and almost half of the sample declares a family income of around EUR 2500/month.

**Table 3.** Descriptive statistics of the two groups (Tasting_g and No_Tasting_g).

| | Tasting_g | | No_Tasting_g | |
|---|---|---|---|---|
| **Continuous Variables** | **Mean** | **St. Dev.** | **Mean** | **St. Dev.** |
| *Age* | 25.08 | $\pm$3.59 | 23.08 | $\pm$2.67 |
| *Family members* | 3.79 | $\pm$1.14 | 3.84 | $\pm$0.88 |
| **Binary variables** | **N** | **%** | **N** | **%** |
| *Gender (male)* | 58 | 43.94% | 74 | 56.06% |
| *Education:* | | | | |
| - *Compulsory* | 0 | 0.00% | 0 | 0.00% |
| - *Upper school* | 75 | 56.82% | 102 | 77.27% |
| - *Bachelor or Master Degree* | 57 | 43.18% | 30 | 22.73% |
| *Income:* | | | | |
| - *Lower than EUR 2500/month* | 26 | 19.70% | 13 | 9.85% |
| - *EUR 2500/month* | 70 | 53.03% | 75 | 56.82% |
| - *Higher than EUR 2500/month* | 36 | 27.27% | 44 | 33.33% |

Table 4 shows that pomegranate is not a widely consumed fruit in the sample: half of the consumers have never consumed the pomegranate in the last year, or consumed it only once (30%), while only 10% claimed to consume it frequently. Regarding the product type, nearly half of the sample stated that, if they could choose, they would prefer to buy pre-packed arils instead of the whole pomegranate. Additionally, half of respondents (50%) prefer to buy pomegranate at the supermarket, while only 14% prefer to buy directly from the producer. Moreover, although more than 80% of consumers believe that it is important to buy fruits from their own regions, only 30% believe that domestic products are more controlled than the imported ones (i.e., score equal to or higher than 7).

**Table 4.** Descriptions of sample consumption habits.

| | | Freq. | |
|---|---|---|---|
| **Survey Statements** | | **N** | **%** |
| 1. | *How often have you consumed pomegranates over the last year?* | | |
| | Never | 56 | 21.21% |
| | Only once | 80 | 30.30% |
| | Several times a year but less than once a month | 101 | 38.26% |
| | At least once a month | 14 | 5.30% |
| | Several times a month | 6 | 2.27% |
| | Several times a month but less than once a week | 4 | 1.52% |
| | At least once a week | 3 | 1.14% |
| 2. | *If you could choose between a whole pomegranate and a package of ready-to-eat arils at the same price, what would you choose to buy?* | | |
| | Whole pomegranate | 159 | 60.23% |
| | Ready-to-eat arils | 105 | 39.77% |
| 3. | *How do you usually consume pomegranates?* | | |
| | Fresh product (whole fruit or arils) | 188 | 71.21% |
| | Pomegranate juice | 48 | 18.18% |
| | Ingredient for recipes | 10 | 3.79% |
| | Other | 18 | 6.82% |
| 4. | *Where do you usually buy fruit?* | | |
| | At the supermarket | 133 | 50.38% |
| | From the specialized shops | 93 | 35.23% |
| | Directly from farmers | 38 | 14.39% |

**Table 4.** *Cont.*

| Survey Statements | Freq. | |
|---|---|---|
| | **N** | **%** |
| **5.** *How often do you buy fruit directly from farmers/farmer cooperatives?* | | |
| Never | 73 | 27.65% |
| Sometimes | 143 | 54.17% |
| Often | 48 | 18.18% |
| **6.** *According to you, is it important that the fruit that you buy comes from your region?* | | |
| Not important at all | 37 | 14.02% |
| Quite important | 142 | 53.79% |
| Very important | 85 | 32.20% |
| | **Mean** | **Std. Dev.** |
| **7.** ***I believe that imported products are less controlled than domestic ones*** (9-point Likert scale item: 1 = totally disagree; 9 = totally agree) | 5.01 | 2.20 |

*3.3. Choice Experiment Estimates*

The model estimates (McFadden pseudo $R^2$ = 0.23) are reported in Table 5. All the fixed parameters are significant at 5%, excepted for *Spec * LL*, *Farmers * LL*, *Farmers * HL*, *WOND_S * LL*, *WOND_V * HL* and *HICAZ * LL* variables. The random parameters are all significant at 5%, except for *farmers* variable, and have low level of heterogeneity, considering that the standard deviation is significant (at the 5% level) only for *WOND_V*, *Spec* and *Local* variables. We assumed that all the random parameters are normally distributed. Price attribute and fixed parameters are considered as fixed [44].

**Table 5.** DCE model results.

| Attribute Level | Coeff. | *p*-Value | WTP |
|---|---|---|---|
| ***Non-Random parameters*** | | | |
| *Price* | −0.21 | 0.00 | |
| *Local * LL* | −1.61 | 0.00 | −7.56 |
| *Local * HL* | −1.13 | 0.00 | −5.32 |
| *Ita * LL* | −1.30 | 0.00 | −6.13 |
| *Ita * HL* | −0.85 | 0.00 | −4.00 |
| *Spec * LL* | −0.37 | 0.31 | |
| *Spec * HL* | −0.86 | 0.00 | 4.03 |
| *Farmers * LL* | 0.34 | 0.39 | |
| *Farmers * HL* | −0.36 | 0.23 | |
| *WOND_S * LL* | 0.31 | 0.66 | |
| *WOND_S * HL* | 1.33 | 0.00 | 6.25 |
| *WOND_V * LL* | −1.74 | 0.00 | −8.21 |
| *WOND_V * HL* | −0.14 | 0.64 | |
| *HICAZ * LL* | −0.86 | 0.05 | −4.04 |
| *HICAZ * HL* | 0.41 | 0.11 | |
| ***Random parameter*** | | | |
| *WOND_S* | −0.85 | 0.00 | −4.02 |
| *WOND_V* | −0.35 | 0.04 | −1.63 |
| *Spec* | 1.33 | 0.00 | 6.27 |
| *Farmers* | 1.33 | 0.40 | |
| *Ita* | 2.33 | 0.00 | 10.99 |
| *Local* | 2.54 | 0.00 | 11.94 |

**Table 5.** *Cont.*

| Attribute Level | Coeff. | *p*-Value | WTP |
|---|---|---|---|
| *Derived standard deviation of random parameter distribution* | | | |
| *WOND_S* | 0.05 | 0.88 | |
| *WOND_V* | −0.86 | 0.02 | |
| *Spec* | −0.90 | 0.06 | |
| *Farmers* | 0.09 | 0.89 | |
| *Ita* | 0.09 | 0.76 | |
| *Local* | 0.96 | 0.02 | |
| Number of respondents | 264 | | |
| Number of Obs | 6336 | | |
| Log-likelihood | −1783.0973 | | |
| McFadden pseudo $R^2$ | 0.23 | | |

Note. HL refers to consumers of the tasting group who gave an overall liking score higher than five to the product sample tasted. LL refers consumers of the tasting group who gave an overall liking score equal or lower than five to the product sample tasted.

The reference baseline is the *HICAZ* variety for the no tasting group, sold in a supermarket, with foreign origin. The first part of the model (non-random parameters) reports the consumers' marginal WTP for the respondents in the tasting group. The second part of the model (random parameters) reports the attributed WTP for consumers in the no tasting group.

For the no tasting group, we find that, on average, consumers prefer to buy *HICAZ* (i.e., the variety that presents the best compromise between color and size) instead of *WOND_V* and *WOND_S* ($\beta_{WOND\_V} = -0.35$ and $\beta_{WOND\_S} = -0.85$). Moreover, they are willing to pay EUR 1.63 and EUR 4.02 less for 100 g of *WOND_V* and *WOND_S* arils, respectively, compared to the *HICAZ* ones. According to Gadže et al. [45] and to Zaouay et al. [46], the visual features that are more appreciated by consumers are large size and red color of the arils.

However, from the magnitude of the standard deviation ($\sigma_{WOND\_V} = 0.86$) it emerged that, although 65% of consumers prefer to buy *HICAZ* instead of *WOND_V*, there is a consistent part of consumers (34%) who preferred *WOND_V* to *HICAZ*. These results are given by $100 \cdot \Phi\left(-\frac{b_k}{s_k}\right)$, where $\Phi$ is the cumulative standard normal distribution and $b_k$ and $s_k$ are, respectively, the mean and the standard deviation of the $k^{th}$ coefficient [47].

Moreover, consumers in the no tasting group prefer the specialized shops sales channel ($\beta_{Spec} = 1.33$) to the supermarket, with a $WTP_{Spec}$ of EUR 6.27/100 g of arils. However, from the magnitude of the standard deviation ($\sigma_{FRUTTIV} = 0.90$) it emerged that 7% of consumers prefer the supermarket sales channel to the specialized shops. Both the Italian ($\beta_{Ita} = 2.33$) and local ($\beta_{local} = 2.54$) origin are preferred to the abroad one. Indeed, consumers are willing to pay EUR 10.99 more for a pack of arils of Italian origin and EUR 11.94 more for local arils, with respect to a 100 g pack of imported arils.

For respondents in the tasting group, we observe that consumers who gave low overall liking scores (LL) to the *HICAZ* sample are less prone to buy it ($\beta_{HICAZ * LL} = -0.86$) than consumers that did not taste the product, thus underling the role of taste and consumers own preferences. Indeed, they are willing to pay EUR 4.04 less than consumers that did not taste the product for a pack of 100 g of *HICAZ* arils. Furthermore, it emerged that *WOND_S* sample is preferred over *HICAZ* (not tasted) for the consumers who like it ($\beta_{WOND\_S * HL} = 1.33$). This is also reflected in the higher WTP for *WOND_S* ($WTP_{WOND\_S * HL}$ = EUR 6.25/100 g of arils) than for untasted *HICAZ* arils. This could be explained by the sample features of WOND_S, generally more appreciated by consumers, and confirms what found in the literature on consumer preferences for pomegranate: the sweetness is the leading factor affecting consumers acceptance for this fruit [46–48]. Sourness and astringency, on the other hand, are negatively correlated with consumers' acceptance: pomegranate and juices perceived as sour [49] and astringent [50] are not appreciated by consumers.

With respect to the reputational attributes, it can be assumed that they are generally less important for the tasted product than for the untasted one. The consumers' WTP for buying products of local ($WTP_{local * LL}$ = EUR $-1.61$/100 g of arils and $WTP_{local * HL}$ = EUR $-7.56$/100 g of arils) or Italian ($WTP_{Ita * LL}$ = EUR $-1.31$/100 g of arils and $WTP_{Ita * HL}$ = EUR $-6.13$/100 g of arils) origin is lower for consumers that tasted the products than for those that did not taste them. Indeed, the magnitude of the difference in consumers' WTP is $-44.6\%$ for local attribute, when the product is highly liked (*Local * HL*) and $-63.3\%$ if the product is less appreciated (*Local * LL*).

To highlight the differences between the tasting and no tasting groups, Table 6 presents the pair comparison of the model estimates and the *p*-value of the F-statistic. For each variable, comparisons were performed between:

- No tasting group and tasting group with High Level (*HL*) of product appreciation;
- No tasting group and tasting group with Low Level (*LL*) of product appreciation;
- Within the tasting group, between *HL* and *LL*.

**Table 6.** Pair comparison of model estimates.

| Variable | Non-Tasting Group | Tasting Group | Between Non-Tasting /Tasting Group | Within Tasting Group |
|---|---|---|---|---|
| | Coeff. ($\beta_i$) | Coeff. ($\beta_j$) | *p*-Value $\beta_i \neq \beta_j$ | *p*-Value $\beta_{HL} \neq \beta_{LL}$ |
| **Origin** | *Local* 2.54 | *Local * HL* $-1.13$ | *** | n.s. |
| | *Local* 2.54 | *Local * LL* $-1.61$ | *** | |
| | *Ita* 2.33 | *Ita * HL* $-0.85$ | *** | n.s. |
| | *Ita* 2.33 | *Ita * LL* $-1.30$ | *** | |
| **Sales channel** | *Spec* 1.33 | *Spec * HL* $-0.86$ | *** | n.s. |
| | *Spec* 1.33 | *Spec * LL* [1] $-0.37$ | *** | |
| | *Farmers* [1] 1.33 | *Farmers * HL* [1] $-0.36$ | ** | * |
| | *Farmers* [1] 1.33 | *Farmers * LL* [1] 0.34 | *** | |
| **Sample** | *Wond_S* $-0.85$ | *Wond_S * HL* 1.33 | *** | n.s. |
| | *Wond_S* $-0.85$ | *Wond_S * LL* [1] 0.31 | ** | |
| | *Wond_V* $-0.35$ | *Wond_V * HL* [1] $-0.14$ | n.s. | *** |
| | *Wond_V* $-0.35$ | *Wond_V * LL* $-1.74$ | *** | |
| | *Hicaz * HL* [1] 0.41 | *Hicaz * LL* $-0.86$ | | *** |

Note. *** $p < 0.01$; ** $p < 0.05$; * $p < 0.1$; n.s. not significant. [1] Parameter not significant in the model estimates (Table 5).

For each comparison, the null hypothesis of the F-statistic is that $\beta_i$ is equal to $\beta_j$, where $\beta_i$ and $\beta_j$ are the estimated parameters of the model associated to the variables object of the comparison *i* and *j* (Table 5). We found that reputational attributes are statistically different between the tasting group and the no tasting group. Specifically, for the origin variable, it emerged that both the local and the Italian origin of the product are perceived as more important for consumers in the no tasting group, compared to the tasting group

(both *HL* and *LL*). No significant difference is found between the *HL* and *LL* group for this variable. Regarding the sales channel variable, it emerged that consumers in the no tasting group gain a higher utility in purchasing pomegranate from specialized shops than consumers in the tasting group (*HL* and *LL*). Thus, in line with Baba et al. [23], the results highlight that the reputation attributes are more relevant during the purchasing decisions for those consumers that can only visually evaluate the product. As broadly demonstrated in the literature, the Italian origin of a food product recalls to a well-defined and universally recognized concept of high quality [51]. Moreover, the literature stresses the fact that consumers perceive products of home-country-of-origin (as Italy is in our case) as being of higher quality than those imported [52]. We also found the same effect for the *local* origin attribute, in line with Tempesta and Vecchiato [53] for milk market, and Sanjuán-López and Resano-Ezcaray [54] for saffron. Despite this, our study highlights that these attributes are subordinate to taste, when consumers experienced the product. Our results are coherent with Torquati et al. [23], who found that liking and tasting are positively correlated with intrinsic cues (i.e., sample), and confirm that reputation attributes are less important than experience cues (e.g., taste), in the purchasing decision [24] for novel food products.

The same goes for the sales channel variable. In our study, consumers attached a greater utility in purchasing pomegranate arils in specialized shops (i.e., greengrocery) when they cannot taste the product, while this sale channel loses importance when they can taste the pomegranate arils. However, it is worth noting that we did not find the same effect for the *farmers* variable. This can be partly explained by the fact that *farmers* is not an attribute with a clear connotation. Especially in sight unseen purchases, this attribute may not be important. In fact, the literature shows that the buying directly from the farmer becomes important when a relationship of trust is established between farmers and consumers [55] and for those consumers who generally prefer to have a direct contact with the producer [56].

## 4. Conclusions

From this research, it is possible to deduce that perceived sensory features significantly explain consumers' purchasing choices, with some differences between tasters and no tasters, and interact with reputational cues. Moreover, reputational attributes (e.g., origin and sales channel) play a stronger role in purchasing choices for no tasters. In fact, both the local and the Italian origin of the product are perceived as more important for consumers in the no tasting group, and the WTP for local/Italian origin is higher for no tasters, compared to the tasting group. However, after the tasting experience, these reputational attributes lose importance (e.g., −44.6% for a product of local origin that is highly appreciated by consumers). Meanwhile, other quality attributes (linked to the sensory profile) gain value, due to the reduced information asymmetry. As an important implication of our study, we can affirm that, if consumers can collect positive consumption experiences for NFPs, this opens them up to building a new reputational framework that is independent of the traditional consumption patterns. Following this, we can conclude that the selling potential of pomegranate arils can even be promising in Italy, where a deeply rooted culinary tradition exists.

Finally, a limitation of this paper is that it refers solely to young consumers, thus preventing the generalizability of the results. However, results suggest some interesting insights about future trends for the demand of NFPs.

**Author Contributions:** Conceptualization. S.T.; methodology, A.S., S.T. and E.G.; statistical analysis, A.S.; investigation, A.S. and E.R.; data curation, A.S. and E.R.; writing—original draft preparation, A.S.; writing—review, S.T. and E.G.; project administration, S.T.; funding acquisition, S.T. All authors have read and agreed to the published version of the manuscript.

**Funding:** This research is linked to the VA_MO Project "Valore Aggiunto Melograno", financed by the Veneto Region's Rural Development Program 2014–2020, measure 16.2.1, linked with EIP-Agri Network.

**Institutional Review Board Statement:** The study was conducted according to the guidelines of the Declaration of Helsinki. Ethical review and approval were waived for this study, because, in accordance with the Declaration of Helsinki, no personal information were collected, data are completely anonymous and the survey do not upset or disturb the participant. Moreover, vulnerable or dependent group are not including in the survey.

**Informed Consent Statement:** Informed consent was obtained from all subjects involved in the study.

**Acknowledgments:** We acknowledge the Agromania cooperative of producers (located in Portogruaro, VE, Italy) for supplying the pomegranate samples produced in Veneto (Italy).

**Conflicts of Interest:** The authors declare no conflict of interest. The funders had no role in the design of the study; in the collection, analyses, or interpretation of data; in the writing of the manuscript, or in the decision to publish the results.

## Abbreviations

| | |
|---|---|
| NFPs | New food products |
| ISTAT | Italian Institute of Statistics |
| DCE | Discrete choice experiment |
| HL | High liking (consumers of the tasting group who gave an overall liking score higher than five to the product sample tasted) |
| LL | Low liking (consumers of the tasting group who gave an overall liking score equal or lower than five to the product sample tasted) |
| QDA | Quantitative descriptive analysis |
| RPL | Random parameter logit |
| WOND_S | Pomegranate of Wonderful variety produced in Sicily (Italy) |
| WOND_V | Pomegranate of Wonderful variety produced in Veneto (Italy) |
| HICAZ | Pomegranate of Hicaz variety produced in Turkey |

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
