# Peer review of "Taste Beats Reputation in New Food Products Choice: The Case of Ready-to-Eat Pomegranate among Young Consumers in Veneto Region (Italy)"

_horticulturae, doi:10.3390/horticulturae7070179_

Round 1

Reviewer 1 Report

Horticulturae 1227803

The above paper examines the consumption of pomegranate arils in Italy.  Specifically, the researchers asked consumers if they favour local producers over foreign producers.  They also determined whether tasting the arils altered their preference for local or foreign products.

The experimental work appears sound.  However, I did have a problem with the overall scope of the study which seems rather superficial.  The paper runs to 20 pages and is very detailed, even though the work only asks a single question: do consumers prefer local products over foreign products if they have the opportunity to taste the arils first?

I suggest that the authors go through the paper and reduce some of the un-necessary texts and analyses.  There are more than 70 references which seems too many to me.

The ‘Introduction’ runs to 2.5 pages and has too many details.  I suggest that this section be reduced to about a single page.  Provide a short overview of pomegranate production and marketing, including the possible preference for local or imported products.  Then indicate the major objective of the study and the use of discrete choice experimentation.

Provide specific details of the main results (including the results of the statistical analyses) in the ‘Abstract’.  It is not readily apparent if consumers prefer local or foreign arils and whether this preference is influenced by previous exposure to the fruit.

There is an overload of information and detail in the ‘Methods’ which runs to five pages.  The methods used by the authors are standard.   Can some of these details be replaced by a few key references?

The information in Figure 1 might be better presented in a Table.

I couldn’t see the value of Figure 2.

There seems to be a lot of unnecessary detail in the ‘Results’ section.  A significant amount of the text could be deleted.  The reader can readily check the main results in the tables and figures.

The information in Figure 3 might be better presented in a Table.

I recommend that the ‘Results’ and ‘Discussion’ sections be combined.  The authors have only asked one question so there is no need for a lot of discussion.

The ‘Conclusion’ should be three to four sentences highlighting the main findings of the study.

I have included a recent overview of world pomegranate marketing (see attached).

To summarize, the current study is rather superficial.  I recommend that the authors focus of a few key issues and reduce much of the unnecessary details.  Twenty pages of text is excessive.

The ‘Title’ should indicate the main findings of the study.

Author Response

We acknowledge the anonymous reviewers for their useful and precious suggestions that enabled us to greatly improve the quality of our manuscript. Accordingly, we completely revised the original paper just following your suggestions. The main revisions of the manuscript have reported in the text with track change.

In the attached file you can find our point-by-point responses to your comments.

Reviewer 2 Report

My observations are essentially related to the setting of sensory analysis.

Line 245 - The propensity to consume new products can be estimated by measuring food neophobia.I would have evaluated this parameter in order to balance the groups of subjects based on the values ​​of food neophobia.What do the authors think about it?  

Line 299 - Can the authors provide more details about the sensory analysis procedure? 

Line 303 - Why were two different scales used? How were the sensory attributes used selected?  

Line 407 - Are men and women equally distributed in the two subgroups?

Author Response

Accordingly, we completely revised the original paper just following your suggestions. The main revisions of the manuscript have reported in the text with track change.

In the attached file you can find our point-by-point responses to your comments.

Reviewer 3 Report

Introduction, line 58 - 67: the health virtues of pomegranate are presented. I wonder if the famous Israelian/English chef Yotam Ottolenghi is as popular in Italy as he is in The Netherlands. His cookery books must have greatly contributed to the popularity of pomergranate arils in my country. So the culinary virtues of the arils might deserve some attention, or doesn't this apply to the Italian market? 

Line 112: what do you mean by the word 'overtime'?

Line 117: as follows

Line 131:'although'  instead of 'despite'

Line 135: involves

Line 141: wine is a product

Line 143/4: wine (possesses, has, demonstrates) a set of different characteristics

Line 145: not: 'a large amount of cue' but ' a large number of cues'. Amount is used for sums of money. And cues should be plural. 

Although your English is quite good and makes agreeable reading, you have rather a lot of mistakes in singular/plural and in the forms of the verbs. I stop mentioning them individually from line 145 onward, and I recommend some further critical reading (that's what I mean by moderate English changes)

Not a 'fully characterization' but a 'full' (still better: complete). Mind also your adverbial forms!

Line 193 - 195: in light of this, in Rozzanigo et al. [44] the three  pomegranate samples were characterise performing both chemical analysis and panel test  to better describe the product and to avoid bias. This sentence is not completely clear to me. Do you mean: 'This has led Rozzzanigo et al. (44) to characterise their pomegranate samples by both chemical analysis and panel tests, to create a better description and to avoid bias.'

Author Response

(The authors gave the same response as above.)

Round 2

Reviewer 1 Report

Horticulturae 1227803 R1

The above paper has been revised by the authors and they provide significant comment to the suggestions provided by the referees.  However, the revised manuscript is not suitable for publication at this stage.

As previously detailed, the manuscript is quite a large production, even though it only asks one major question.  Most sections of the paper are excessive, including the ‘Abstract’.  I suggest that the authors go through the paper carefully and delete the information that is not important.

The ‘Abstract’ can probably be reduced by half.  The authors still do not present the results of any of the statistical analyses, especially those answering the main question.

The ‘Introduction’ and ‘Background’ should be combined and reduced to no more than a single page.

The ‘Conclusion’ should be reduced to just three to four sentences summarizing the main findings of the study.

To summarize, the authors’ case would be strengthened with a better focus on the main findings of the study.  The large body of text reduces rather than enhances the science and commercial significance of the work.

Author Response

We acknowledge the anonymous reviewer for the useful and precious suggestions. Accordingly, we further revised the original paper following the suggestions. The main revisions of the manuscript have reported in the text with track change.

In the following pages, you can find our point-by-point responses to your comments.

The above paper has been revised by the authors and they provide significant comment to the suggestions provided by the referees.  However, the revised manuscript is not suitable for publication at this stage.

As previously detailed, the manuscript is quite a large production, even though it only asks one major question.  Most sections of the paper are excessive, including the ‘Abstract’.  I suggest that the authors go through the paper carefully and delete the information that is not important.

The ‘Abstract’ can probably be reduced by half.  The authors still do not present the results of any of the statistical analyses, especially those answering the main question.

Accordingly, the abstract has been reduced.

The ‘Introduction’ and ‘Background’ should be combined and reduced to no more than a single page.

The introduction and the background have been merged and reduced.

The ‘Conclusion’ should be reduced to just three to four sentences summarizing the main findings of the study.

Accordingly, the conclusion has been further reduced.

To summarize, the authors’ case would be strengthened with a better focus on the main findings of the study.  The large body of text reduces rather than enhances the science and commercial significance of the work.

Thank you again for the suggestions.

Round 3

Reviewer 1 Report

Horticulturae 1262078

The revised paper is much improved.  I wasn’t sure if the text highlighted in yellow was to be deleted.  I would like to review the paper without the text highlighted.

A few other comments.

Indicate the meaning of “WTP” in the Abstract.

Line 21. Delete “New Food Products (NFPs) (i.e.”

Line 25. Delete “as those here investigated”

Add a note to the ‘Abstract’ indicating that the results of the study may not be applicable to the general (older) population.

Could some data relating to the quality evaluation be added to the ‘Abtract’?

The study is too limited to add significantly to a broad view of discrete choice experiments in horticulture.

Author Response

We acknowledge the anonymous reviewers for their useful and precious suggestions that greatly improve the quality of our manuscript. Accordingly, we completely revised the original paper just following your suggestions. The main revisions of the manuscript have been reported in the text with track change.

In the attached file you can find our point-by-point responses to your comments.
